# Distinguishing representational geometries with controversial stimuli: Bayesian experimental design and its application to face dissimilarity judgments

**Tal Golan**[*]
Zuckerman Mind Brain Behavior Institute
Columbia University
New York, NY 10027
`tal.golan@columbia.edu`

**Wenxuan Guo**[*]
Department of Psychology
Columbia University
New York, NY 10027
`w.guo@columbia.edu`

**Heiko H. Schütt**
Zuckerman Mind Brain Behavior Institute
Columbia University
New York, NY 10027
`hs3110@columbia.edu`

**Nikolaus Kriegeskorte**
Departments of Psychology, Neuroscience, and Electrical Engineering
Columbia University
New York, NY 10027
`n.kriegeskorte@columbia.edu`

## Abstract

Comparing representations of complex stimuli in neural network layers to human brain representations or behavioral judgments can guide model development. However, even qualitatively distinct neural network models often predict similar representational geometries of typical stimulus sets. We propose a Bayesian experimental design approach to synthesizing stimulus sets for adjudicating among representational models efficiently. We apply our method to discriminate among candidate neural network models of behavioral face dissimilarity judgments. Our results indicate that a neural network trained to invert a 3D-face-model graphics renderer is more human-aligned than the same architecture trained on identification, classification, or autoencoding. Our proposed stimulus synthesis objective is generally applicable to designing experiments to be analyzed by representational similarity analysis for model comparison.

Neural network models have become increasingly important in cognitive computational neuroscience to avoid the ambiguity of verbal hypotheses and quantitatively capture behavior and neural activity. As our representational models improve, we encounter a new challenge. Models of human perceptual representation are typically evaluated with respect to natural or naturalistic stimulus sets (e.g., photographs or rendered images of 3D objects) chosen by the experimenter. However, there is no guarantee that the stimulus set we choose will reveal the distinct representational geometries in the models we wish to compare. Several recent studies find that data sets are explained equally well by multiple qualitatively distinct models. For example, in a study employing natural images as

---

[*]The first two authors contributed equally to the work.

4th Workshop on Shared Visual Representations in Human and Machine Visual Intelligence (SVRHM) at the Neural Information Processing Systems (NeurIPS) conference 2022. New Orleans.

stimuli, multiple neural network architectures showed similar level of correspondence with human neural representations [1]. The challenge of similar representational geometries predicted by distinct models is even more pronounced in the subdomain of face representation, where natural face stimuli span a relatively limited image manifold. Jozwik et al. [2] found that several qualitatively distinct computational models predicted human behavioral face dissimilarity judgments almost perfectly. Equivalent accuracy levels of distinct face representation models were also observed when the model dissimilarities were compared to neural representational dissimilarities measured by fMRI [3] or ECoG [4]. While the above studies used representational similarity analysis [RSA 5] to compare human and model representations, the problem of non-discriminable predictions of distinct models is not unique to RSA. Model discrimination is expected to be even more challenging in encoding analyses, where the many parameters of the fitted linear mapping render each model more flexible and reduce model discriminability even in ideal, noiseless conditions [6, 7].

One way forward is to test model-brain alignment with controversial stimuli: stimuli generated to induce distinct predictions in different models [8–10]. These studies optimized the controversiality of the stimuli in terms of the image labels or relative sentence probabilities assigned by different models, reflecting the human behavioral tasks used to adjudicate among the models (see also Maximum Differentiation Competition, [11, 12]). Here, we put forward a stimulus-synthesis method that generates stimulus sets for which different models predict distinct *representational geometries*. Our proposed method can compare fixed models by their unweighted representational geometries (i.e., "fixed RSA", see [13]) or flexible models whose weighting parameters have already been estimated. We apply the proposed method to discriminate among a set of neural network models of behavioral judgments of face similarity. The proposed optimization objective for stimulus synthesis is applicable for designing RSA experiments in general, beyond the domain of faces, and for experiments that measure either brain activity or behavioral judgments.

# 1 Methods

## 1.1 Stimulus optimization objective

In order to generate a stimulus set that supports efficient model discrimination, we take a Bayesian Optimal Experimental Design (BOED) approach [14, 15]. Unlike typical BOED applications where the "design" consists of experimental protocol parameters such as stimulus presentation order and timing, here we optimize the experimental stimulus set itself.[2]

**Differentiable model discrimination objective.** To generate a stimulus set that efficiently discriminates between $M$ representational models, we would ideally want each potential data-generating model to be clearly distinct from *all* other models. We propose to use each model in turn as the data-generating model and maximize a global utility $U(\xi)$ that measures the expected advantage of the data-generating model over the best-performing alternative model:

$$U(\xi) = \sum_m p(m) \cdot f\Big( \underbrace{\psi(\tilde{\mathbf{y}}^m, \hat{\mathbf{y}}^m \mid \xi)}_{\substack{\text{how well model } m \text{ predicts} \\ \text{data generated by a held-out} \\ \text{instance of itself}}} - \underbrace{\max_{m' \neq m} \psi(\tilde{\mathbf{y}}^m, \hat{\mathbf{y}}^{m'} \mid \xi)}_{\substack{\text{how well the best other model} \\ \text{predicts data generated by} \\ \text{model } m}} \Big), \quad (1)$$

where $p(m)$ is our prior belief in model $m$ to be the best candidate model (a uniform distribution if we have not collected any data yet), $\tilde{\mathbf{y}}^m$ is the vector of dissimilarities predicted by model $m$ for stimulus set $\xi$, $\hat{\mathbf{y}}^m$ is the corresponding prediction vector made by another instance of model $m$ with different weight initialization, $\psi$ is a model-performance estimator (see below) and $f$ is a monotonically increasing function, which we set here to emphasize negative differences (i.e., incorrect model recoveries) and de-emphasize positive differences (i.e., correct model recoveries). Here we used the saturating function $f(x) = -e^{-10x}$, which led to a high rate of recovery of the true data-generating model in simulations (where we know the data-generating model) compared to alternative choices for $f(x)$, including the identity function.

**Model performance estimator.** The model-performance estimator $\psi$ should be as close as possible to how the human-model alignment will be evaluated in the analyses of the actual data, but needs

---

[2]We think of our controversial stimulus sets as "optimized" rather than "optimal" since we generate them by means of local optimization, which does not guarantee a globally optimal solution.

to be differentiable to allow efficient stimulus optimization. In the case of neuroimaging RSA studies, each model is evaluated by its representational dissimilarity predictions across all stimulus pairs, yielding a representational dissimilarity matrix (RDM). Such RDMs can also be estimated from behavioral judgments [16]. To estimate model performance based on RDMs, whitened RDM correlation measures are appropriate [17]. For estimating model performance based on dissimilarities between independent stimulus pairs (where each stimulus appears only in one pair), simple correlation coefficients are usually adequate.

**Accounting for multiple neural network layers.** In studies that evaluate deep neural networks as computational models of human representational geometries, multiple layers of each neural network must be considered as potential representational models. A common approach is to evaluate each neural network by its most human-aligned layer (e.g., [18, 1, 19]). In principle, we can simply treat multiple layers as multiple models in the framework proposed above. However, such an approach might devote a large portion of the design's power to discriminating among the consecutive layers of single models (which tend to have related representations). Our main goal, however, is to adjudicate among entire neural network models. We therefore introduce layers as an unknown nuisance parameter in the experiment, yielding a slightly more elaborate optimization objective:

$$
U(\xi) = \sum_m p(m) \cdot \sum_{l \in \{1,\ldots,L_m\}} p(l \mid m) \cdot f\Big( \max_{l'} \psi\big(\tilde{\mathbf{y}}^{m,l}(\xi), \hat{\mathbf{y}}^{m,l'}(\xi)\big) - \max_{m' \neq m} \max_{l'} \psi\big(\tilde{\mathbf{y}}^{m,l}(\xi), \hat{\mathbf{y}}^{m',l'}(\xi)\big)\Big),
$$

(2)

where $p(l \mid m)$ is our prior belief that layer $l$ is the data generating representation given model $m$, and $\tilde{\mathbf{y}}^{m,l}(\xi_t)$ and $\hat{\mathbf{y}}^{m,l}(\xi_t)$ are the vectors of dissimilarities predicted by this layer, according to two different instances. We refer the instances marked by hat (ˆ) as "reference" instances and reuse them in data analysis.

## 1.2 Optimizing face stimulus sets for discriminating among models of face dissimilarity judgments

We applied the approach described above to design controversial stimulus sets for a behavioral experiment that compares neural network models of face dissimilarity judgments.

We trained six neural networks sharing the veteran VGG-16 architecture [20] on different datasets and objectives (Table 1, Appendix A.1).

| architecture | training task | training dataset |
|---|---|---|
| VGG-16 | object categorization | ImageNet [21] (object photographs) |
| | face identification | VGGFace2 [22] (face photographs) |
| | face identification | BFM (synthetic faces) |
| | autoencoding (VAE) | VGGFace2 (face photographs) |
| | autoencoding (VAE) | BFM (synthetic faces) |
| | inverse rendering | BFM (synthetic faces) |

Table 1: Six candidate neural network models of human face representation.

We used the 3D morphable Basel Face Model (BFM 2019; [23]) as our face stimulus generator. This generative model parameterizes individual face stimuli using separate latent spaces for shape, expression, and texture (see Appendix A.2). The latents define 3D face structure and texture independently from identity-invariant parameters such as pose and illumination. In a previous study, Daube et al. [24] used a generative face model to synthesize faces that are similar to a reference individual according to a neural-network-based encoding model (Fig. 5 in [24]). This was done as a means of testing each candidate representational model. Here, the generative face model constrains the synthesis of sets of faces designed to best discriminate among alternative representational models.

We considered an experimental paradigm in which in every trial, participants arrange $N$ face pairs along a dissimilarity axis [2] (Fig. S1). Model performance (i.e., how well model $m$ predicts the response set $y$ to stimulus set $\xi$) is quantified by first measuring the correlation between model dissimilarity predictions and the participant responses within each trial, and then averaging the

resulting correlation coefficients across $T$ trials:[3]

$$\psi(y, m \mid \xi) = \frac{1}{T} \sum_t \text{corr}\left(\mathbf{y}_t, \hat{\mathbf{y}}^m(\xi_t)\right),\qquad(3)$$

where $\mathbf{y}_t$ is the vector of observed dissimilarity judgments in trial $t$ and $\mathbf{y}^m(\xi_t)$ is the corresponding vector of dissimilarities predicted by model $m$. We used the squared Euclidean distance between the stimulus representations within each pair as the predicted dissimilarity. $\text{corr}(\cdot, \cdot)$ is a correlation coefficient. We used Pearson's linear correlation coefficient for stimulus optimization and Spearman's rank correlation coefficient[4] for analyzing the human experiment (the former is differentiable, whereas the latter is invariant to order-preserving transformations). We constructed three multi-trial stimulus sets of BFM synthetic faces:

**Random stimulus set.** As a baseline condition, we randomly sampled 144 faces from the BFM distribution (i.e., $\mathcal{N}(0, I)$) and rendered them as a frontal view. We randomly allocated the faces to pairs and assigned six face pairs to each of the 12 trials. Here, as well as in all other conditions, the BFM expression component was turned off, so variation was limited to the shape and texture components.

**Systematically sampled stimulus set.** For this stimulus set, we systematically sampled face pairs with different BFM distances. This condition aims to induce a large amount of explainable variability in the across-subject mean pattern of dissimilarity judgments, as often informally done by experimenters. Following random-sampling and pairing, we constrained the BFM shape and texture latent vectors of each of 144 faces to have an $l_\infty$-norm $\leq 2.0$ (i.e., reside in a hyper-cube extending $\pm 2$ SDs from the mean face in each dimension). We then adjusted the BFM shape and texture latents such that for each trial, one pair would have a zero Euclidean distance, one pair would have a maximal distance and occupy opposite corners of the hyper-cube, and the remaining four would evenly sample the range of distances in between (but not necessarily lie on the same line in face space). This was achieved by iterative constrained optimization, minimizing the squared deviations of the pairs' distances from the pre-determined distances.

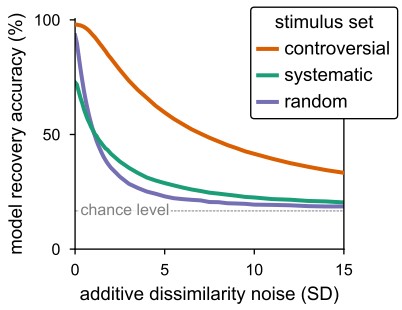

Figure 1: Simulated model recovery accuracy for three stimulus sets: (1) randomly sampled; (2) systematically sampled; and (3) controversial. One held-out instance per model was used to simulate observed dissimilarity vectors, assuming a uniform prior over the six models and 16 layers. The simulated dissimilarities were compared to the dissimilarities of the set of instances used as references in the stimulus optimization procedure (see Appendix A.4). To simulate measurement noise, Gaussian noise was added to each representation (x-axis). The noise was scaled such that SD=1 would correspond to the SD of the representation's noiseless dissimilarities in the randomly sampled stimulus set. The y-axis denotes the percentage of simulations (n=2000) in which the data-generating model (regardless of the data-generating layer) was recovered.

**Controversial stimulus set.** Here, we used the same initialization and $l_\infty$-norm constraints but optimized model discriminability. We maximized Eq. 2 (plugging in Eq. 3) by iteratively adjusting the faces' latents while monitoring the representation of the resulting 2D face image in each layer of the six candidate models (evaluating two instances per model). We implemented a differentiable approximation to the BFM rendering process in PyTorch3D [26] to enable backpropagation of the gradients from the neural network representations through the 2D image to the BFM latents. This enabled us to use SGD optimization for stimulus synthesis (see further details in Appendix A.3).

## 1.3 Human behavioral experiment

We recruited 90 US-based, 18-60 year-old participants (39 female, 32.5±10.4 years mean age) through the `prolific.co` platform (Appendix A.5). Our experimental procedures were approved by the Columbia University Institutional Review Board (protocol number IRB-AAAR9520). Each stimulus set (random, systematic, and controversial) was tested on an independent set of 30 participants. The

---

[3]This is a variation on [2], where the dissimilarities were first concatenated across trials and only then correlated with the model predictions, yielding a single correlation coefficient.

[4]Ties were handled by an analytical random-among-equals tie-breaking estimate ([25], Eq. 14).

participants were presented with two repetitions of $T = 12$ unique trials in a randomized order. In each trial, the participants were instructed to arrange six face pairs according to their perceived dissimilarity along a vertical dissimilarity axis, labeled "identical" and "maximally different" on opposite ends (Figs. S1 and S2). The arrangement task was implemented on the Meadows web platform (`meadows-research.com`). Unlike the design in [2], no visual examples of identical and maximally different face pairs were provided. Each generated face image appeared only in one pair.

To obtain an unbiased estimate of each model's best layer performance, we evaluated each subject-model correlation using the model layer that best predicted the other 29 subjects' ratings (bottom panels in Fig. 2). A lower bound on the best possible model performance for each data set was calculated by predicting each participant's responses by the average response vector of the other 29 participants [27], calculated after rank-transforming the six ratings within each trial. An upper bound was calculated in the same fashion while including the predicted participant's response vector in the across-subject average.

## 2  Results

**Simulations.**  We first considered the discriminability of the six models for each of the three stimulus sets in silico (Fig. 1). We simulated model recovery experiments by comparing the dissimilarities predicted by each model and each layer to the dissimilarities predicted by held-out model instances. This was done both in a noiseless setting and under Gaussian noise added to the simulated dissimilar-

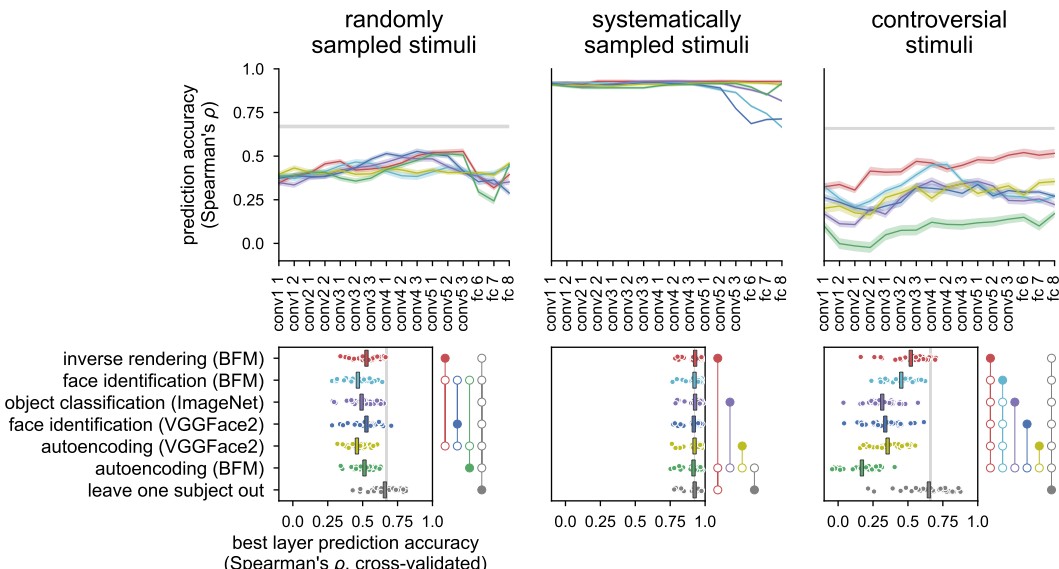

Figure 2: **Prediction accuracy of human dissimilarity judgments of six VGG-16 neural networks trained on different tasks and datasets**. Each model is evaluated on a randomly sampled stimulus set (left column), a systematically sampled stimulus set (middle column), and a controversial stimulus set (right column). **Top row:** Each panel shows the average Spearman's $\rho$ between the dissimilarity vectors of each model layer and human response for one stimulus set. Each colored line indicates performance of one model, and the shaded areas represent the SE. The grey regions indicate noise ceiling estimates on the average model-human alignment an ideal model can achieve, given the between-subject variability in human responses. **Bottom row:** Cross-validated performance estimates of each model's best layer. Each dot represents the rank correlation between one subject and one model, using the layer that best explained the other subjects' responses. Significance indicators: A solid dot connected to a set of open dots indicates that the model aligned with the solid dot has significantly higher correlation with the human judgments than any of the models aligned with the open dots ($p < .05$, paired t-tests on Fisher-transformed correlation coefficients, Holm-Šídák FWE-corrected for 21 comparisons).

ities. As expected, the controversial stimulus set enabled more accurate model recovery, especially under noise. See Figure S3 for an ablation study of the optimization algorithm.

**Behavioral results.** Both randomly sampled and systematically sampled face pairs failed to successfully adjudicate among the six candidate models: no model was significantly more accurate than all of the others in predicting the human responses. In contrast, when models and humans were evaluated on the controversial stimulus set, the neural network trained on inverse rendering (i.e., predicting BFM latents from the input image) was found to be significantly more human-consistent than the five other models. We replicated this result with a second controversial stimulus set, optimized from a different random initialization of face latents (Fig. S4). Note that the systematically sampled stimulus set, which was designed to elicit maximally distinct levels of perceptual dissimilarity, indeed yielded highly reliable human judgments. Below the resulting high noise ceiling, however, the performance differences among the alternative models were small and largely insignificant. All models came close to the noise ceiling, illustrating how apparent good model performance can fail to drive theoretical progress.

Controversial stimuli provide a "magnifying glass" for model differences [28]. We therefore consider the prediction accuracy for a controversial stimulus set as a test statistic for model comparison rather than as an absolute benchmark of the models.

## 3    Discussion

In this work, we put forward a controversial stimulus synthesis procedure for RSA experiments and applied it to compare models of face representation. Beyond faces, the proposed optimization approach can be readily applied to design controversial stimulus sets for differentiable representational models including neural networks, in vision and in other representation domains.

**Limitations and future directions.** Using gradient-based stimulus optimization enables efficient search and convergence in high-dimensional stimulus spaces (here, $\mathbb{R}^{398}$). However, gradient-based stimulus optimization requires differentiability of the objective, the stimulus renderer, and the models. Adapting evolutionary algorithm-based stimulus synthesis [29] to directly maximize model discrimination may allow a complementary approach. See also [30] for random-sampling-based and [31] for evolutionary-algorithm-based approaches to stimulus selection.

**Implications for modeling human face representations.** We found that a neural network trained to invert the graphics rendering process and estimate the BFM latent representations that gave rise to a face image yielded the most human-aligned representational geometry. This result is consistent with the finding of Yildirim et al. [32] obtained using macaque intracranial recordings. However, it should be noted that this network was tested in favorable conditions since it was trained on inverting the rendering process of the generative model used to parameterize the experimental stimuli. In future work, we will evaluate the models' performance using controversial stimuli parameterized by alternative generators (e.g., generative adversarial networks) to address this limitation.

## Acknowledgments and Disclosure of Funding

This publication was made possible with the support of the Charles H. Revson Foundation to TG. The statements made and views expressed, however, are solely the responsibility of the authors. This material is based upon work supported by the National Science Foundation under Grant No. 1948004 to NK. We acknowledge Dr. T. Vetter, Department of Computer Science, and the University of Basel, for the Basel Face Model data.

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

# A    Appendix

## A.1    Detailed description of model training

All of our models (Table 1) had the VGG16 architecture [20] as implemented in torchvision and were trained on $128 \times 128$ pixel input images. Neural network weights were initialized by randomly sampling from a zero-mean Gaussian distribution, as described in [33]. No batch normalization was employed. We used PyTorch Lightning for model training (`www.pytorchlightning.ai`). All the models were trained using four GeForce RTX 2080 Ti GPUs. We trained three instances for each of the six models described below, starting from different random weight initializations. The first instance was used as a representational model ($\hat{\mathbf{y}}^m$ in our equations). The second instance was used to simulate ground-truth responses during stimulus optimization ($\tilde{\mathbf{y}}^m$), and the third was used to simulate ground-truth responses in the model recovery experiments. For the sake of brevity, the validation performance described below is for the first instance. Drop-out was used during training, but disabled during inference (i.e., when testing the neural networks on held-out data or using them as representational models).

Each network was trained on one of three data sets: ImageNet [21], VGGFace [22], or synthetic faces we sampled and rendered from the Basel Face Model. For each data set, the mean and the standard deviation of the pixel intensity levels were estimated for the three color components of each image and then averaged across a sample of the data set's training images. These estimates were used to shift and scale the color channels of the input image to ensure expected zero mean and unit standard deviation inputs during training. The same intensity normalization was applied when the neural networks were used in stimulus optimization and data analysis.

**Object recognition (ImageNet).**    We trained a VGG16 network on a 1000-way object classification task using the ImageNet data set. The images were preprocessed and augmented using datamodules.ImagenetDataModule, which is implemented in Lightning Bolts [34]. The training hyperparameters were as in the pretrained torchvision model reference script (`https://github.com/pytorch/vision/tree/main/references/classification`). We trained the network using stochastic gradient descent (SGD) with a weight decay of 0.0001, momentum of 0.9, and mini-batches of 1,024 images, minimizing the cross-entropy loss. The learning rate was initialized to 0.1 and decreased by a factor of 10 every 30 epochs. During both training and validation, each image was rescaled such that the shorter side was 146 pixels and then resized to $128 \times 128$ pixels using a random crop during training and a centered crop during validation. During training, the images were also augmented by random horizontal flip, applied with a probability of 0.2. The network was trained for 90 epochs and achieved 1.56 validation loss and 62.4% top-1 and 84.3% top-5 validation accuracy.

**Face identification (VGGFace2).**    We trained a network of the same VGG-16 architecture to classify 8,631 face identities in the VGGFace2 dataset, minimizing the cross-entropy loss. Images were cropped using the data set's predefined bounding boxes and then rescaled and cropped again (random cropping during training and centered cropping during validation). Additionally, training images were augmented with greyscale transformation with a probability of 0.2. For each identity, we held 10% of the images from the training data for validation. We used SGD with a weight decay of 0.0001, momentum of 0.9, and mini-batches of 1,024 images. The learning rate was initialized to 0.01 and decreased by a factor of 10 every ten epochs. The network was trained for 30 epochs and achieved 0.310 validation loss and 95.7% top-1 and 98.2% top-5 validation accuracy.

**Face identification (BFM).** A third VGG-16 network was trained to identify 8,631 synthetic face identities (matching the number of identities in the VGGFace2 natural face photo data set). Here the synthetic face images were generated with the Basel Face Model. For each synthetic identity, all of the face images shared the same randomly sampled shape and texture latents but had different expression latents, pose, lighting direction, and lighting intensity. Within-identity variability was further increased with 16 naturalistic augmentations [35], including different forms of noise addition, cutout, blurring, and grayscale transformation. 363 images of each identity were generated to roughly match the total number of training images in the VGGFace2 dataset [22]. The rendered images were preprocessed similarly to our VGGFace2 face identification model, with random cropping during training and centered cropping during validation. The training (minimizing the cross-entropy loss) was carried out by SGD with a weight decay of 0.0001, momentum of 0.9, and mini-batches of 512 images. The learning rate was initialized to 0.01 and reduced by a factor of 10 every ten epochs. The model was trained for 30 epochs and reached a 0.00169 validation loss and 99.95% top-1 and 99.99% top-5 classification accuracy for the validation set.

**Autoencoding (VGGFace2).** We trained a variational autoencoder (VAE, [36, 37]) on the VG-GFace2 dataset. The autoencoder consisted of a VGG-16 as an encoder and an "upconvolutional" decoder. The penultimate layer of the VGG-16 was mapped by two fully connected layers to a predicted mean of a 500-dimensional Gaussian latent representation and to a variance parameter for each dimension, defining a diagonal covariance matrix. During training, the 500-dimensional latent vector was sampled according to the predicted mean and covariance and was then transformed back to an image by the decoder. The decoder network mapped the latent from its 500-dimensional space to a stack of 256 $8 \times 8$ activation maps. These maps were then upsampled by four trainable upconvolutional blocks. Each block doubled the size of the maps and halved the number of channels. Each block included a $3 \times 3$ convolutional layer followed by a PixelShuffle upsampling operation, ReLU non-linearity, an additional $3 \times 3$ convolutional layer that maintained the activation dimensionality, and a second ReLU non-linearity. After the last block, the resulting 16 $128 \times 128$ maps were mapped by a $3 \times 3$ convolutional layer and a sigmoid non-linearity to the original RGB image format. The VAE was trained to minimize the evidence lower bound (ELBO). The $\beta$ hyper-parameter (the trade-off between reconstruction error and prior regularization) was set to 1.0 and the $\sigma$ hyper-parameter (the scaling of the reconstruction error) was empirically fitted during training to minimize the ELBO [i.e., we used an "$\sigma$-VAE", 37]. We trained the network with ADAM [38] with $\beta_0 = 0.9$, $\beta_1 = 0.999$, $\epsilon = 10^{-4}$, a weight decay of 0.00001, and mini-batches of 256 images. The learning rate was initialized to 0.0005 and was reduced by a factor of 10 after 20 epochs. After training for 30 epochs, the network achieved 0.679 MSE for validation images. Visually, the reconstructed validation images had high fidelity. After training was completed, we discarded the decoder and used only the VGG-16 encoder as our representational model. For the ultimate layer, we used only the predicted mean of the latent representation without sampling from the predicted Gaussian in latent space.

**Autoencoding (BFM).** We repeated the VAE training procedure with the dataset of BFM synthetic faces described in the section Face identification (BFM) above. After training for 30 epochs, the network achieved 0.554 MSE for validation images. Here as well, the reconstructions had high visual fidelity.

**Inverse rendering (BFM).** Inspired by the Efficient Inverse Graphics (EIG) model by Yildirim et al. [32], we trained a VGG-16 to predict BFM face latents and extrinsic properties–pose and lighting. In the EIG model, different intermediate layers were trained to fit different representational targets, ranging from image segmentation to identity. Here, we trained a neural network to map each input image to a single output vector consisting of six components: 199 shape latents, 199 texture latents, 100 expression latents, four face pose parameters (expressed as a quaternion), three lighting color and three lighting direction parameters. This target representation resembles the EIG's $f_5$ representation, which was found by [32] to best match the macaque anterior face patch AM. We generated 3,300,000 unique synthetic faces using BFM by randomly sampling face latents, pose, lighting color, and direction. The loss function was defined as the sum of the normalized mean squared error (NMSE) for the six components. To maintain the veracity of the training targets, the only data augmentation during training was random cropping. We trained the network with ADAM [38] with $\beta_0 = 0.9$, $\beta_1 = 0.999$, $\epsilon = 10^{-8}$, no weight decay, and mini-batches of 512 images. The learning rate was initialized to 0.0001 and was reduced by a factor of 10 every 40 epochs. After training for 120 epochs, the model achieved 2.31 NMSE loss for validation images (where 0.0 is perfect performance and

6.0 is the error of the best constant predictor). Feeding the inferred latent representations back to the BFM renderer generated face images highly consistent with the input images.

## A.2 Basel Face Model

The Basel Face Model (BFM) was conceived as a PCA-based probabilistic graphics model of faces [39] that combines 3D shape and mapped texture components to render nearly photorealistic images of human faces. The BFM characterizes each face by a shape, a texture, and (optionally) an expression, using separate latent vectors for each of these three determinants of face appearance. The shape of a face is parameterized by a latent vector $\boldsymbol{\alpha}$, defined such that $\mathbf{s}(\boldsymbol{\alpha}) = \bar{\mathbf{s}} + \sum_{i=1}^{r} \alpha_i \sqrt{\lambda_i} \mathbf{u}_i$ , where $\mathbf{s}(\boldsymbol{\alpha})$ is a vector of vertex coordinates defining a 3D face mesh, $\bar{\mathbf{s}}$ is the vector of vertex coordinates of the average face mesh, $\lambda_i$ is the i-th shape eigenvalue, and $\mathbf{u}_i$ is the i-th shape eigenvector. Texture components are handled analogously, enabling general linear mixing of 3D face models. The shape and texture components define a normal distribution in latent space, which in the 2017 version was based on 3D scans of 200 people (not randomly sampled to be representative of any particular population). The normal distribution model enables us to sample faces at random by drawing the parameters $\alpha_i \sim \mathcal{N}(0, 1)$ or to evaluate the likelihood of a face with respect to that distribution. Note, however, that this distribution is not representative of the human population. In our optimization procedure, we adjust the latent vectors for both the shape and texture models. See [23] for a detailed mathematical description of the more recent BFM versions, where the PCA was generalized to a truncated Karhunen–Loève expansion over face deformations.

## A.3 Stimulus optimization procedure

The synthesized stimuli were initialized as shape and texture BFM latent vectors randomly sampled from the BFM normal distribution model and then projected into a hyper-cube subtending $\pm$ 2 SDs. The face latents were then iteratively adjusted to maximize the optimization objective (Eq. 2).

Note that the stimulus optimization objective in Eq. 2 considers two trained model instances per model architecture and objective. This approach serves to safeguard against confusion of random variability across instances of a model trained from different random seeds for variability between different model architectures and objectives. For conceptual clarity, we defined one instance for each model as the "reference instance", denoting its predictions as $\hat{\mathbf{y}}^m$. This instance for each model was also used in data analysis. The other instance was used as the ground-truth model during the stimulus optimization, and its predictions are denoted by $\tilde{\mathbf{y}}^m$. The optimization objective in Eq. 2 considers only correlations *between* the reference and ground-truth instances. Note, however, that our stimulus-synthesis procedure can be improved by also considering the correlations between distinct models within each instance set. Furthermore, the minimal setup of two instances per model can be extended to a larger instance sample at the cost of greater GPU compute requirements.

We used the Adam optimizer with a learning rate set to 0.01. In this context, the learning rate controls the speed of face latent adjustment. In each optimization iteration, we evaluated the optimization objective ten times with jittered versions of the stimuli. We used a differentiable scale jitter between 99.5% and 100.5%, followed by a differentiable translation jitter of up to 10 pixels. After averaging the objective across these ten presentations, we took an optimization step ascending the gradient. Reparameterization kept the optimized face latents within the hyper-cube constraint: we parameterized each face latent dimension $\alpha_i$ by $\alpha_i = 2 \tanh(\alpha_i')$, optimizing $\boldsymbol{\alpha}$ and using $\boldsymbol{\alpha}$, which lies in $(-2, 2)$, to generate faces. The learning rate was reduced by a factor of 2 after apparent convergence. Apparent convergence was defined as no significant improvement of the mean objective in the last 20 iterations compared to the mean objective in the preceding 20 iterations. Learning rate reduction was carried out twice. After the third apparent convergence (or the 1000-th iteration, whichever happened first), the optimization was terminated.

We used a compute server with eight GeForce RTX 2080 Ti GPUs. Two GPUs were devoted to PyTorch3D image rendering and the other six to model evaluation (we evaluated 144 images on 12 neural networks: 2 instances $\times$ 6 models). We used activation checkpointing to reduce GPU RAM requirements.

### A.4 Model-recovery experiments

For noiseless simulations (the datapoints on the left edge of Fig. 1), model-recovery accuracy was estimated by

$$A(\xi) = \sum_m p(m) \cdot \sum_{l \in \{1, \dots, L_m\}} p(l \mid m) \cdot \mathbb{1}\Big[ \max_{l'} \psi\big(\tilde{\mathbf{y}}^{m,l}(\xi), \hat{\mathbf{y}}^{m,l'}(\xi)\big) > \max_{m' \neq m} \max_{l'} \psi\big(\tilde{\mathbf{y}}^{m,l}(\xi), \hat{\mathbf{y}}^{m',l'}(\xi)\big) \Big],$$
(4)

using uniform priors $p(m)$ and $p(l \mid m)$. The reference dissimilarities $\hat{\mathbf{y}}^{m,l'}(\xi)$ were obtained from the same reference-model instances as used in the stimulus optimization. Ground-truth dissimilarities ($\tilde{\mathbf{y}}^{m,l}(\xi)$) were obtained from held-out instances.

This proportion measures how often the ground-truth representational dissimilarities are better predicted by the ground-truth model rather than by any of the other models. It is estimated on average across all possible ground-truth models and layers.

For simulations including measurement noise (i.e., the rest of the data points in Fig. 1), the ground-truth representational dissimilarities were first normalized by the mean representational dissimilarity of the same model and layer presented with the random stimulus set. We then added independent standard Gaussian noise to each ground-truth representational dissimilarity before comparing them to the reference representational dissimilarities.

### A.5 Human experiment screening criteria and participation compensation

**Screening criteria.** We took several steps to ensure that the participants recruited through prolific.co understood the arrangement task and made a sincere effort. A participant was excluded from the data analysis if they failed to meet any of the three criteria:

- Successful completion of a practice trial. In the practice trial, the participants arranged image pairs of various objects. We tested whether each participant's arrangement conformed to all of the following dissimilarity relations:
    1. (dog, strawberry) > (apple, orange)
    2. (car, apple) > (apple, orange)
    3. (apple, orange) > (apple, apple)
    4. (apple, orange) > (strawberry, strawberry).

- Low within-subject reliability. Since each participant was presented with every trial twice, we could estimate within-subject reliability by correlating the dissimilarity ratings in the two repetitions, yielding one (Speamran's $\tau$) correlation coefficient per trial. For each participant, we tested the significance of the Fisher-transformed correlation coefficients by a one-sample t-test against zero. Participants with $p \geq .05$ were disqualified for low reliability.

- Short participation duration. Participants who spent less than 10 minutes doing the experiment were excluded from analyses.

**Participation compensation.** We paid $816 in total to 136 participants, including 30 replication study participants and 16 disqualified participants who failed one the screen criteria. Three participants failed more than one screening criteria and were not paid. The hourly payment was $15.4, on average.

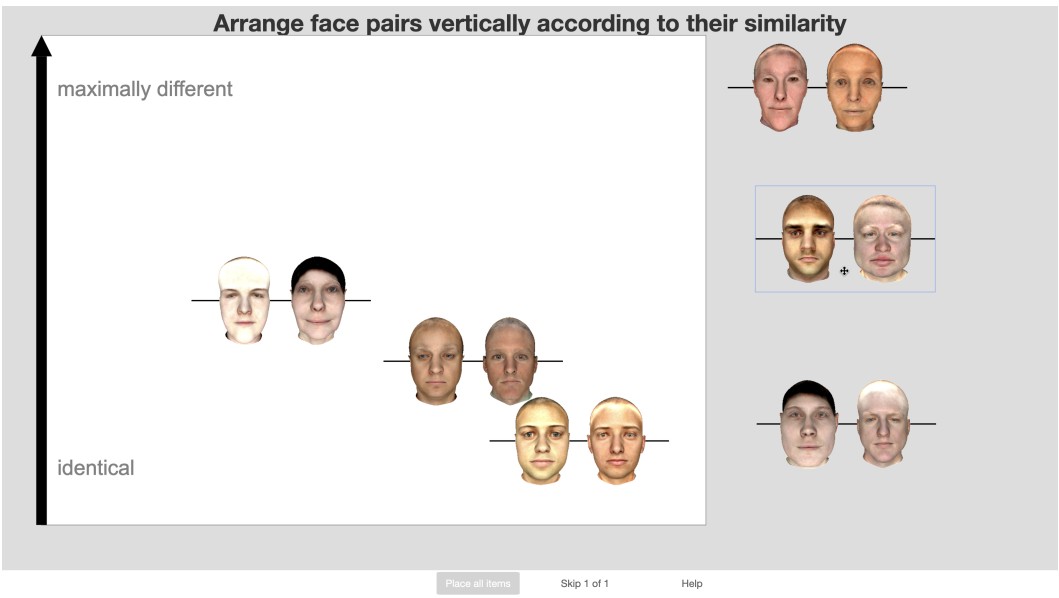

Figure S1: **An example experimental trial.** After completing an object-pair arrangement task for practice, participants were shown an annotated illustration of the face-pair arrangement task, with the following instruction alongside:

> In the following task, you will arrange face pairs vertically according to their similarity. In each trial, each pair of faces that are relatively more dissimilar from each other should be placed above each pair of faces that is relatively more similar to each other. Note that only the vertical positioning of faces will be taken into account. Horizontal space is provided so you can arrange the face pairs more easily.

For each arrangement screen, participants dragged each face pair from the grey area to the white arena and arranged the pairs vertically to indicate their perceptual similarity.

**Note that the high prevalence of Caucasian faces reflects the particular sample underlying the Basel Face Model.** Using alternative generators (or modified BFM latent distributions) can enable tuning the ethnic composition of the faces to better fit particular human populations.

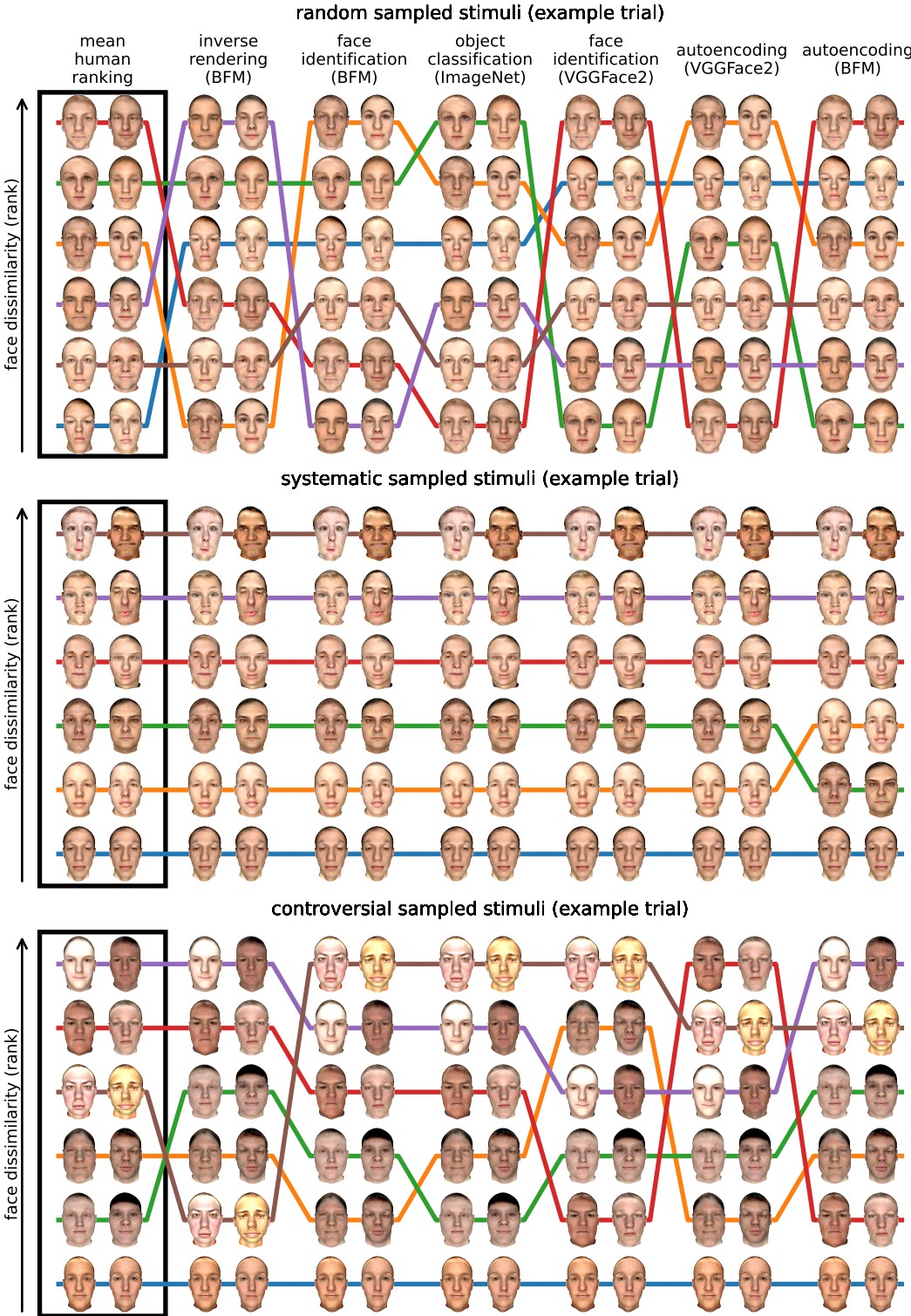

Figure S2: **Stimuli and dissimilarity rankings in one randomly sampled trial from each stimulus set: random, systematic and controversial stimuli.** Each colored line indicates the dissimilarity ranking of one face pair. The average human ranking is shown on the left (the ranked average across all participants' ranked dissimilarity ratings). The ranked predictions of each of the neural network models are shown on the right. For each model, we show here the predictions of its best-performing layer (see Fig. 2, top row), which is not necessarily the one that best predicts these particular trials.

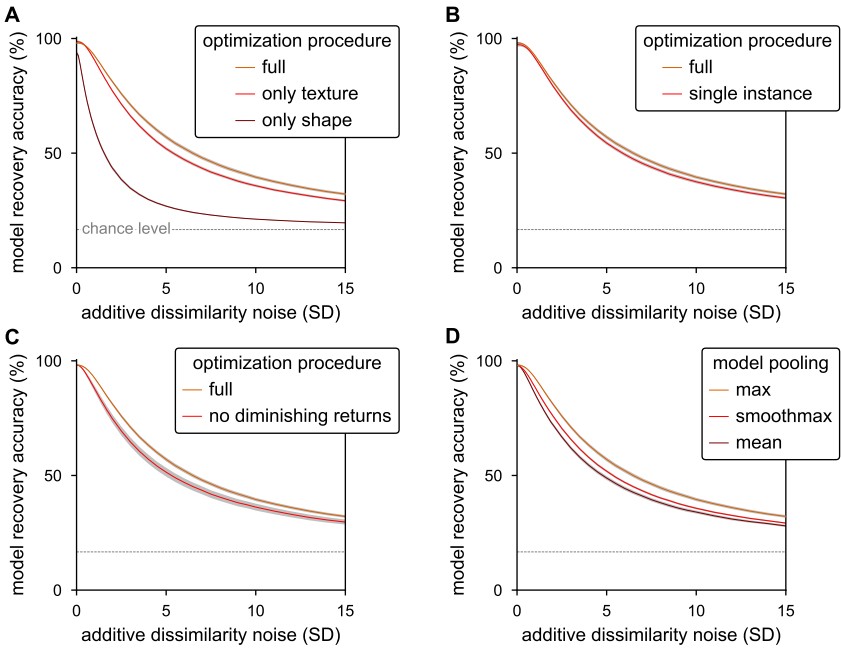

Figure S3: **Stimulus optimization ablation study.** We generated controversial stimulus sets either with the full optimization procedure described in the main text (orange-brown) or with various components turned off (red hues). We simulated model recovery accuracy for each stimulus set (see Fig. 1) and averaged the results across 10 stimulus sets of each optimization condition (each stimulus set was generated from a different random initialization of the face stimuli). The curves depict the mean model recovery accuracy as a function of simulated noise. The gray shaded regions depict $\pm 1$ SE of this measure. **(A)** Optimizing only the texture component of the faces while keeping the randomly sampled shape component fixed leads to a slightly reduced model recovery accuracy compared to optimizing both components. Optimizing only the shape component leads to a dramatic reduction in model recovery accuracy. **(B)** Using only a single instance of each neural network model leads to a small reduction in model recovery accuracy. In this setting, the first term in Eq. 2 is constant, and the optimization focuses on features that are more idiosyncratic to the particular weight initializations of the neural network instances used (see [40]). **(C)** Setting the function $f(\cdot)$ in Eq. 2 to identity (i.e., $f(x) = x$ instead of $f(x) = -e^{-10x}$) reduces model recovery accuracy and increases the variability of this measure across repeated stimulus optimization runs. When $f(x)$ is set to identity, there are no diminishing returns for further separating a pair of representations that are already well-separated compared to other representation pairs. **(D)** The effect of alternative model-performance pooling functions. "smoothmax": replacing the maximum over models in Eq. 2 with a smooth maximum, defined as $\mathcal{S}_\alpha(\mathbf{x}) = \sum_i x_i \cdot e^{\alpha \cdot x_i} / \sum_i e^{\alpha \cdot x_i}$. Here $\alpha$ was set to 10.0. "mean": replacing the maximum over models with an average over models. "max": a hard maximum over alternative models, as employed in our behavioral experiment. The hard maximum yielded the highest model-recovery accuracy, even though it is not a smooth function.

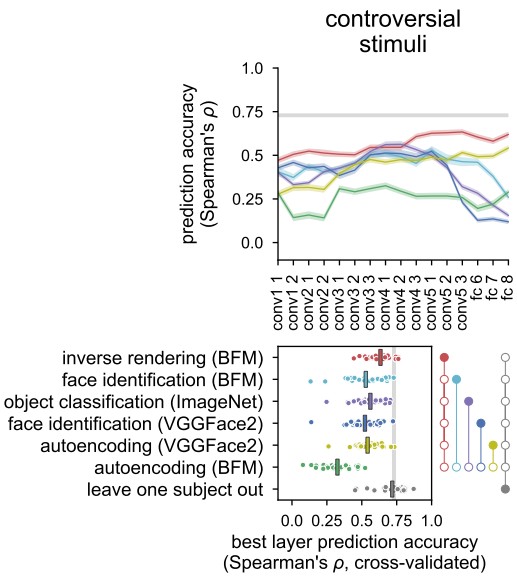

Figure S4: **A replication of the controversial stimuli experiment using a different controversial stimulus set.** Conventions are as in Figure 2. Since the controversial stimulus set was a product of random initialization followed by local optimization, we assessed the replicability of the results arising from using such stimulus sets by testing another group of 30 subjects with another controversial set, initialized with a different random seed. We found a result pattern consistent with the first controversial stimulus set we tested (Fig. 2, right column). Importantly, the neural network trained to invert the rendering of 2D face images performed significantly better than all other models in this replication.

