# OpenReview forum: "Distinguishing representational geometries with controversial stimuli: Bayesian experimental design and its application to face dissimilarity judgments"
_NeurIPS.cc/2022/Workshop/SVRHM — SVRHM Oral_

### Official Review · Reviewer_ZNu3 · 2022-10-13
**Outstanding work**

**Rating:** 10
**Confidence:** 5

**Review:**

This paper develops a framework for generating stimuli for which the representational geometries of different models will maximally differ (controversial stimuli). They then compared representational geometries between models (from layerwise activations) and humans (from dissimilarity ratings) for faces that were either randomly sampled, systematically sampled, or from this controversial stimulus set. All models predicted human behavior for randomly sampled stimuli and systematically sampled stimuli, but models differed for controversial stimuli, with a model trained on inverse rendering outperforming the best.

The paper is very well written, the work is well motivated (addressing an important need), the approach is highly original and general, the results were compelling, and overall the work is presented very clearly. Developing the framework for generating these stimuli is alone a signifiant contribution to the field, and seeing a convincing demonstration of the power of the approach really pushes the paper over the top from my perspective. I think the challenge of distinguishing between models is fundamental for those interested in using DNN models to understand human perception.

I have a few minor points/questions, which aren’t really critiques, but I think the answers affect the general utility of the approach.

First, how can / should the effect sizes can be contextualized? Suppose two models are nearly perfect models of face perception in humans, but one of the models is better at representing something minor, like eyebrow texture. A controversial stimulus set could tease apart which is the best model (potentially showing a dramatic difference between these models in correspondence to human behavior), but really they are both very good models, and one of them is just a smidge better on one dimension that was exaggerated in the controversial stimulus set. So, intuitively, I would say model 2 is slightly better than model 1 (because model 2 gets eyebrow texture better), but the measured effect might be very large (because controversial stimuli are like a magnifying glass), and I also don’t know how much people really rely on eyebrow texture in general so I don’t know the importance of this difference in the more natural scenario.

Second, could the authors say more about why the stimuli need to be synthesized, rather than mined from a large natural image dataset? Is it simply the case that you wouldn’t find sufficient independent variation along features-of-interest to generate a strongly controversial stimuli? This seems important because there aren’t many domains were we have access to differentiable stimulus renderers, but perhaps that’s just a matter of time.

Finally, and somewhat relatedly, is it a coincidence that the controversial stimuli were generated from a specific face-rendering-generative-model, and that the best performing VGG-16 model was the one that was trained to invert this generative model? It would be reassuring if a dataset mined from real-world images (as opposed to a synthesized dataset) showed converging evidence that the inverse-rendering task results in more human-like representations.

---

### Official Review · Reviewer_JXve · 2022-10-14
**Very nice work**

**Rating:** 9
**Confidence:** 3

**Review:**

In this paper, authors tackle the problem of the difficulty in distinguishing between models for the best fit to human brain or behavioral data, even when these models are very different. The authors use the approach of generating a stimulus set that maximizes the discriminability of the models, or ‘controversial stimuli’.

The problem is important, the approach is very nice, and the results are interesting. I don’t have major concerns about the work, just a couple of thoughts.

One point that I think could be somewhat clearer is that the benefit of this procedure doesn’t really come from cutting out non-controversial stimuli from the training set, but from adding the controversial stimuli. What I mean is that while a set of controversial stimuli is more informative than an equally sized set of non-controversial stimuli (e.g. randomly chosen), I imagine that the combined set is more informative than either alone. So, this method basically maximizes the informativeness of a stimulus set given a fixed size. This is not a critique of the work, since for most (or all) experiments there is a maximum stimulus set size, and so the method is very useful. But I think that it could aid the clarity to mention that this is the issue that experimenters face: they have limited resources that limit the size of the stimulus set they can test, and so they want to maximize the informativeness of this limited-size set. A possible misunderstanding if this is not clear, could be that the distracted reader initially thinks that the method could be used to cut out some of the the stimuli from the stimulus set from an experiment that was already done, while I understand that the method wouldn’t be useful for this. The authors could reasonably disagree with the view that this needs further clarification though.

One conceptual question for the authors about a possible “limitation”, that I don’t have a good intuition on, is whether this method could generate controversies between models by manipulating features that are not what the experimenter is interested in. As an extreme example, lets suppose that the method is applied to a synthesizer of images that look like white noise, using the same models and task analyzed here. I imagine it’s possible that the method would find a set of noise-like stimuli that are quite controversial between the models, but we would likely not learn much about face perception by comparing human behavior and models with these stimuli. Although this extreme example seems uninteresting (human responses would be random), I wonder if less extreme examples are relevant. E.g. could this procedure generate controversy based on low-level feature manipulations that affect the models differently, and end up selecting the model with low-level processing most similar to humans, instead of the most similar in higher-level representations? To clarify, I don’t mean the problem of the model maximizing early-layer dissimilarities, but finding a nuisance variation (e.g. contrast) that affects one networks representations more than the other. I also don’t mean that being a problem for the current stimulus synthesizer, but for the extension of the method to other paradigms. I don’t think this thought requires any concrete action from the authors, but if they have some thoughts of this, and in what cases the method may not be useful (or even be misleading) by such things, it could be an interesting addition to the manuscript.

---

### Official Review · Reviewer_bCpv · 2022-10-16
**a method for distinguishing models of face perception**

**Rating:** 8
**Confidence:** 5

**Review:**

This paper applies the methodology of controversial stimuli introduced by Golan et al. 2020 (closely related to maximum differentiation (MAD) competition introduced by Wang and Simoncelli, 2008) to the domain of face perception. The authors investigate different neural network representations to see which representation aligns with human face perception.

Quality:
The experiments presented in the paper are well thought out, both in terms of stimulus generation and in terms of the human ratings.

Clarity:
The paper is generally clear, although I was a bit confused about the motivation and results for the control using the “systematically sampled” stimulus set. Naively, it seems like this systematically sampled set should do no worse than the random set at distinguishing models, and yet the predictions for this set are nearly all at the maximum prediction performance. Guiding readers through this control and why it is of interest would help.

Originality:
The methodology used in the paper is original, in particular the extension of using RDMs as the measure for separating two models, and also the method of including multiple layers as a nuisance variable.

Significance:
The paper seems generally of interest for studying aspects of face perception and for distinguishing various computational models that work equally well on sampled natural stimuli. However, I was left wondering about larger motivation for the work. Can we learn anything from the generated stimuli to better understand human face perception?  Do we learn anything other than a “ranking” of models which would tell us how to improve models of face perception in the future? The methodology is the most interesting part of the study, but using it to understand aspects of perception would further strengthen the work.

---

### Official Review · Reviewer_DD6R · 2022-10-16
**Excellent paper, interesting problem and very-well explained solution**

**Rating:** 9
**Confidence:** 3

**Review:**

The authors propose a new stimulus design method that highlights differences in internal representations of different models, while accounting for differences due to different initializations of the same model. The method can be used to generate new stimuli which highlight differences in representational geometries of different models. In representational similarity analysis, it is often observed that when using experimenter-selected stimuli, many very-different models can have similar representational geometries. This makes it difficult to compare different models and determine which ones are most human aligned. They use their method to compare VGG networks trained on different tasks and find that models are trained on the task of inverting a 3D-face-model graphics renderer are most aligned to human behavior. This was difficult to do with randomly selected or even systematically chosen stimuli.

The problem is very interesting, significant and should be of interest to many researchers, especially those working with representation similarity analysis. Their approach is very well explained, novel and thorough.

The results and discussion sections could be expanded on to improve the paper. Specifically,  explaining what is model recovery accuracy, expanding on the results of Figure1, and a discussion on how internal representational differences affected task-level decisions of the model would be helpful.

---

### Official Review · Reviewer_EXaD · 2022-10-17
**An innovative approach to controversial stimulus generation, with fairly convincing validation analyses**

**Rating:** 8
**Confidence:** 3

**Review:**

Here the authors introduce a new Bayesian procedure for controversial stimulus generation that is applicable and timely, with potential for broad impact for those who conduct studies comparing models to human brain/behavioral data. This specific method addresses a core challenge facing the field right now, which is that distinct models often predict highly similar representational geometries. Developing an approach to generate controversial stimuli that predict distinct representational geometries, as opposed to image labels or relative sentence probabilities, represents an important point of novelty and may be especially useful for future studies that seek to come up with stimulus sets for fMRI studies of visual representation. The idea motivating the optimization procedure is intuitive: the goal is to generate a stimulus set that maximizes the difference between the performance of a given model as assessed using data generated by a held-out instance of itself, and the performance of the next-best model at predicting that same data. The authors test their method by attempting to discriminate between candidate neural network models of behavioral face similarity judgements, and show that their stimulus generation procedure effectively separates out the performance levels of the candidate models, favoring an inverse rendering model trained to predict Basel Face Model latents.

- Pros
	- The writing is clear and cohesive, and helps the reader to understand the optimization procedure without necessarily requiring a background in Bayesian methods.
   - The models selected for the face recognition analysis nicely span a range of relevant candidates, including object and facial recognition models, autoencoders, and inverse rendering models.
   - Approaches like these have potentially far-reaching implications for how stimuli are selected in future representational experiments that involve models, brains, and/or behavior.
	- The appendices are extensive, providing important insight into each stage of the methodology.
	- All figures contain appropriate noise ceilings that help put model performance levels into perspective.
- Suggestions
	- It is interesting that the model discrimination objective involves comparing model m against only the best-performing of the other candidate models (via the max() operator on the right-hand side of equation 1). This seems like an important choice that contrasts with an optimization procedure that would seek to maximize e.g. the sum of differences between the target and alternative models, or the sum of differences between all pairs of models jointly. It would be useful for the authors to justify this specific choice in the main text, and/or test out alternative approaches to see if they impact the main findings.
	- The human behavioral dataset seems to be relatively low-dimensional (only 12 unique trials per subject, and only six face pairs per trial). This paradigm is sufficient to evaluate the efficacy of the stimulus generation procedure and validate that the controversial stimuli do in fact adjudicate between candidate models, but it is hard to read strongly into the primacy of the inverse rendering BFM model given (a) the scope of the behavioral dataset, (b) the fact that synthetic rather than naturalistic stimuli were used, and (c) the fact that the model was tested in favorable conditions (as noted in the Discussion). Overall, these points do not detract majorly from the overall utility of the method and the value of the paper, given the compelling comparison between randomly-sampled, systematically-sampled, and controversial stimuli shown in the top row of Figure 2.
	- A major question is whether this approach could be used to come up with a dataset of naturalistic (object, scene, or face) images. It would be useful if the authors could elaborate on whether/how the current procedure would need to be modified to do so (beyond alluding to GANs in the final sentence of the main text).

Overall, I feel that this innovative and thorough paper merits acceptance at SVRHM, and it should generate interesting discussion among workshop participants.